# Building expertise through task-specific representational alignment in biological and artificial neural networks

**Ashug Gurijala**
Department of Computer Science
California Institute of Technology
Pasadena, California, 91106
agurijala@caltech.edu

**Michael Nyugen Jr.**
Department of Computer Science
University of California Davis
Davis, California, 95616
michaelnguyenjr0@gmail.com

**John O'Doherty**
Department of Humanities and Social Sciences
California Institute of Technology
Pasadena, California, 91106
jdoherty@caltech.edu

**Sneha Aenugu**
Department of Humanities and Social Sciences
California Institute of Technology
Pasadena, California, 91106
saenugu@caltech.edu

**Editors:** Marco Fumero, Clementine Domine, Zorah Lähner, Irene Cannistraci, Bo Zhao, Alex Williams

## Abstract

Humans can generate both rapid and accurate responses in diverse tasks by building perceptuo-motor expertise through practice. Expert responses are robust to task-irrelevant distractors and state-space nuisance. In this paper, we investigate the representational transformations that guide skill acquisition in both humans and artificial agents. Specifically, we investigate the hypothesis that the evolution of task-specific efficient representational coding emerges in the higher layers of the visuo-motor hierarchy in biological and artificial networks. Towards this end, we built a custom shooter game with the specific aim of introducing maximal variance in perceptual state spaces, in which the development of expertise entails building robustness to such state-space distortions. Deep reinforcement learning agents playing the game develop representational alignment with the task-relevant features in higher layers late in the training process, with the lower layers remaining agnostic to the task. We aim to investigate parallel representational alignment in humans through longitudinal neural recordings to precisely probe the evolution of representational bottlenecks that result in the formation of expertise.

## 1 Introduction

An expert driver can drive effectively in challenging conditions, such as rain or snowstorms. Even in such low-visibility state spaces, scene perception and subsequent maneuvering remain robust.

An expert birdwatcher can identify the correct bird species even when there is significant variation within individual species. Such perceptuomotor skills develop in humans through practice, with experts showing a marked difference in performance compared to novices. Experts may see the world differently from novices; if so, where in the perceptual hierarchy do such differences emerge? What transformations do representations undergo in the shift from novice to expert?

DiCarlo and Cox [2007] argue that populations of neural responses are tuned to create disentangled representations that effectively guide invariant object perception. When perceptual hypotheses are entangled, object perception is rendered difficult, degrading performance. Effective disentanglement of task-relevant variables could facilitate expert responses that are rapid and accurate. If so, where in the perceptuo-motor hierarchy are representations disentangled?

Moreover, excessive task-specific alignment could lead to fragility. Being hyper-focused on one task can degrade performance in other tasks that could impinge on the agent at any time. This creates a trade-off between being task-generalists and task-specificists. As humans are task-generalists and can acquire competence across diverse tasks, representational disentanglement is likely localized to a few layers of the visuo-motor hierarchy, enabling rapid task switching. Mante et al. [2013] demonstrates through neurophysiological evidence and recurrent network modeling that irrelevant sensory inputs are not filtered out in the early stages of the visual hierarchy, but rather pruned away in the integration stage in the prefrontal cortex.

Representational bottlenecks likely emerge at specific layers of biological representations to support both expert action performance and task-switching flexibility. We can likely track information bottlenecks in the brain through examining representational transformations that evolve during skill acquisition. Specifically, quantifying to what extent task-irrelevant signals are suppressed and task-relevant signals are amplified gives us a measure of the representational bottleneck in a layer Tishby et al. [2000], Achille and Soatto [2018]. We can likewise trace representational bottlenecks in the layers of artificial networks. Tacchetti et al. [2017] showed that architectures that most facilitated invariant object recognition were also the ones to capture human neural recordings best.

In this study, we investigate the representational bottlenecks that result in task-specific expertise in humans and artificial agents. Through longitudinal neural recordings from a human training from novice to expert on a perceptuo-motor task, we aim to track representational transformations across the visuo-motor hierarchy in the brain during the process of building expertise and to identify parallels and divergences in various network architectures for building expertise in artificial agents.

## 2 Experiment

We designed a custom 2D shooter game in Unity (see Figure 1). The objective of the game is to shoot as many enemy targets as possible before the level ends. Enemies are split into three categories, and a specific bullet can hit each category. Visual features of the enemy indicate its category, and this needs to be learned through trial and error. Figure 1(b) shows the enemy features and three bullet actions in the game. Enemies' visual features can be tagged with five attributes: head gear, body color, enemy-weapon type, facial expression, and eye color. Unknown to the participants, one feature of the enemy is designated as the target feature, and the correct bullet is mapped to it. In this study, we designated the enemy weapon as the target feature, with three instances: axe, spear, and sword. We mapped each instance to one of three bullet types in the game: lightning, fireball, or iceshard.

The game is thus a visual category-learning task, made challenging by composite enemy features that serve as distractors. Moreover, the game includes a navigation component in which the player must navigate through obstacles in the maze to reach the enemy. There are background non-enemy objects that serve as distractors, in addition to changing ambient conditions such as day, night, rain, and snow. The maze's layout is also divided into four terrains that are visually distinct. Enemies spawn one at a time and stay in the scene for a set duration of time (8-12 seconds) before they disappear if not shot. Participants navigate through the maze to reach the enemy's shooting range and choose the appropriate bullet to shoot the enemy and earn points in the game.

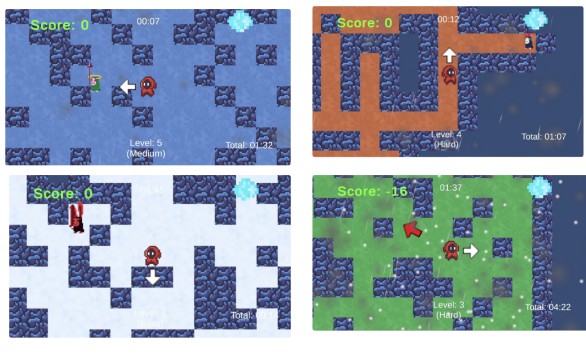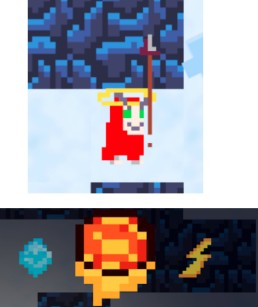

(a) Game scenes in different terrains and weather conditions.

(b) Enemy visual features; One enemy feature is predictive of the bullet category that can be used to hit it.

Figure 1: Shooter game. Player locates the enemy and moves towards it to shoot with a specific bullet. Enemy disappears within a time limit if not shot.

## 3 Humans exhibit incremental and insight-like learning patterns in the task

We recruited five participants aged 19-31 through the online platform Prolific. Each participant played 23 levels in total, each level lasting 2 minutes, with a total game time of 46 minutes.

Figure 2 (top) shows learning patterns of five human participants in the task. Three of the participants successfully learned the enemy-weapon mapping during the task. Participants displayed sudden insight-like learning patterns: upon figuring out the mapping rule, they made consistent, correct responses, in contrast to the episodes before the insight. Different participants achieved insight-like learning at varying stages of the game, with one participant learning the mapping after playing just two levels.

Figure 2 (bottom) plots the time between successive hits during the course of the game. The metric indicates how quickly participants were able to navigate the maze to reach the enemy's shooting range. Although marked by a steep learning curve in the beginning, participants continue to demonstrate steady improvement in performance throughout the game. Four out of five participants show continuous improvement in handling game controls and navigating the obstacles around the maze. One participant showed no learning and, in fact, showed a decline in performance, indicating disengagement.

We dissociated the two subtasks of the game — steady movement-based navigation and sudden insight-like category learning — as two separate learning strategies — procedural (implicit) and declarative (explicit), akin to the two memory systems in the brain Squire [2004]. We modeled separate artificial agents for the movement and shooting subtasks to capture these distinct phases of learning.

## 4 Training produces a gradient of increasing target selectivity in the layers of artificial agents

We trained modular deep reinforcement learning (RL) agents using a proximal policy optimization (PPO) algorithm Schulman et al. [2017]. We split the game into two subtasks — movement and shooting Lample and Chaplot [2016]. The movement subtask navigates through the maze and reaches the enemy, and the shooting subtask learns to map the enemy with the specific bullet type. More details about the training procedure are provided in the supplementary section.

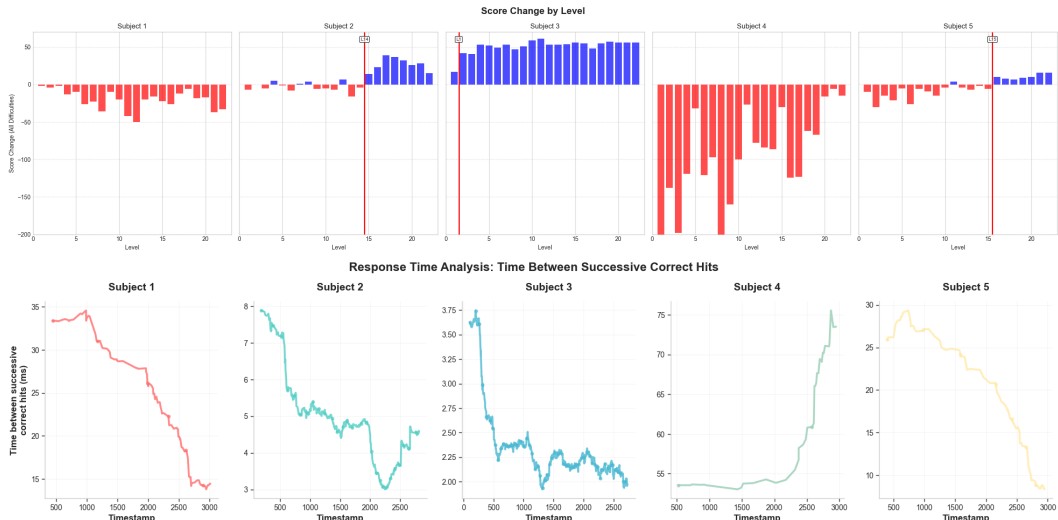

Figure 2: **Top:** Human learning curves showing score change per level, with positive score changes highlighted in blue. The red vertical line indicates the time point where the insight-like learning manifests. **Bottom:** Time between successive hits reduced over the course of the game, indicating improved navigation and game control skills.

We computed the accuracy of predicting the bullet type (action), enemy weapon type (target), and enemy facial expression (distractor) from different layers of the CNN in the shooting agent PPO. Throughout the training, the final layers were tuned to predict the target feature while decreasing their predictability of the distractor variable (Figure 3). Likewise, in the movement agent, the predictability of the biome (terrain type) decreased across the higher layers, as it is irrelevant to the movement action (Figure 4). However, day/night is still highly predictable, as night offers limited visibility and might affect movement. These results align with the hypothesis that representational bottlenecks emerge in higher layers of the network to facilitate task-specific expertise.

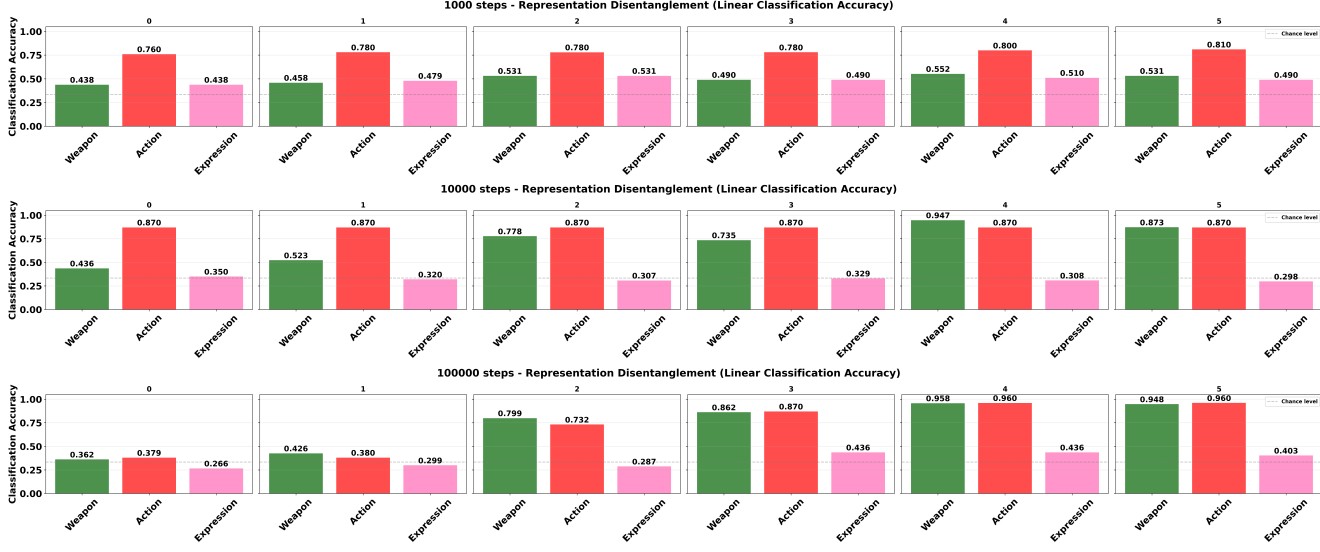

Figure 3: Decoding accuracy of action (bullet type), target (enemy weapon), and distractor (enemy facial expression) from different layers of the CNN in the shooting agent PPO at different stages of training.

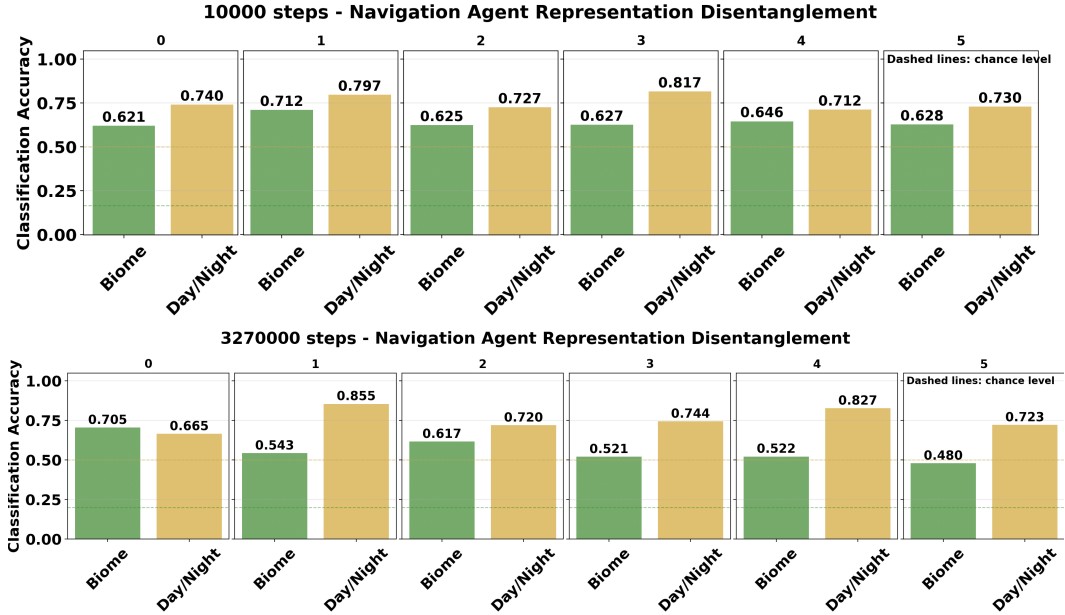

Figure 4: Decoding accuracy of the terrain type (biome) and day/night from different layers of the CNN in the movement agent PPO at early (top) and late (bottom) stages of training.

# 5 Future course of action

Our next steps involve a parallel investigation into human learning to compare with the artificial agent's representational changes. We plan to collect longitudinal functional MRI (fMRI) data from six participants. Each participant will play the game for approximately four hours, split across multiple days. To ensure the task remains dynamic and to probe the flexibility of representational learning, we will shuffle the target feature and introduce new enemy features throughout the experiment. This will allow us to track the evolution of representational transformations within the human brain's visuo-motor hierarchy as expertise is built and as task rules are switched.

We plan to train different AI architectures, specifically ones that are built on unsupervised disentanglement of enemy features for a more human-like priors in representations Higgins et al. [2017]. By doing this, we can investigate whether a more human-like prior over representations leads to skill-acquisition patterns that more closely resemble those of our human participants. This comparative approach will provide a comprehensive understanding of how expertise is formed in both biological and artificial systems.

### Acknowledgments

We thank the O'Doherty lab members for their critical feedback through the course of the project. Ashug Gurijala received support from the Caltech Student-Faculty Programs (SURF). The research was sponsored by the U.S. Army Research Office and accomplished under cooperative agreement W911NF-19-2-0026 for the Institute for Collaborative Biotechnologies.

# A  Supplementary Material

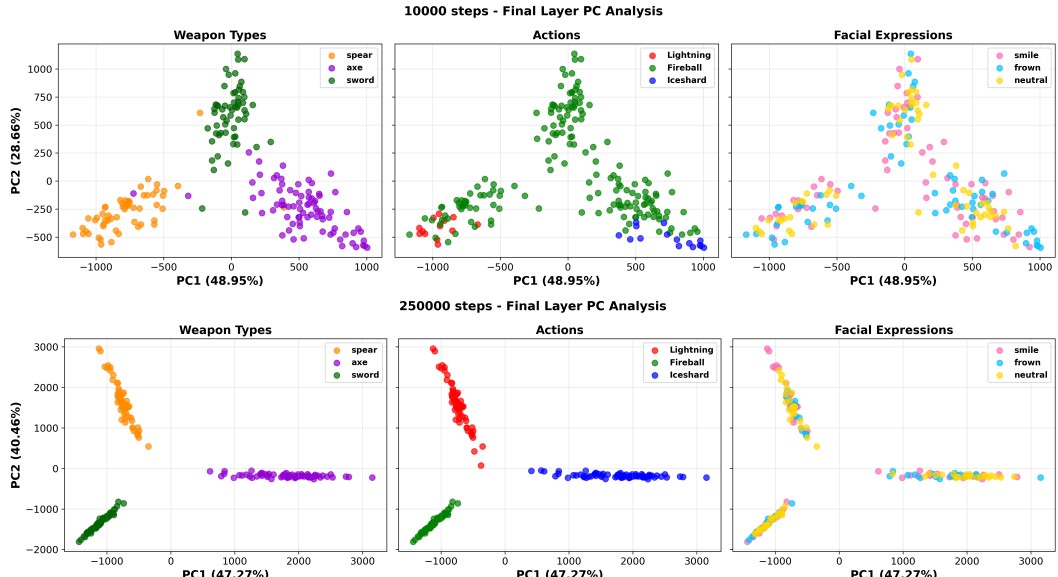

Figure 5: PCA visualization of the final layer activations in the shooting agent PPO at different stages of training. Enemy weapon type is the target feature while facial expression is the distractor. Enemy weapon type representations become more disentangled and aligned with that of action representations through the course of training, while the distractor feature becomes more entangled.

## A.1  Environment

The platform is a 2D Unity game (60 fps) consisting of procedurally generated mazes. They are generated using random-walker generation to ensure that participants cannot rely on spatial memory or fixed navigation patterns. The environment varies with the following:

- **Biomes:** Arctic, Desert, Forest, Volcano, Ocean, and Sunset biomes alter color schemes and visual context.
- **Weather & Time:** Weather effects (Rain, Snow, Fog) and day/night cycles add to visual complexity.
- **Controlled complexity parameters:** Maze shape and connectivity ensure consistent difficulty while maintaining novelty.

## A.2  Task

The task is modeled as a partially observable Markov decision process (POMDP).The core challenge is a dual-objective task that combines maze navigation with feature-based classification.

An observation $o_t \in \Omega$ is an RGB image rendered from a top-down camera in the 2D environment. This camera is fixed to the player's position, keeping them centered within the frame and revealing only a segment of the maze within a fixed radius. The state of the environment, $s_t \in \mathcal{S}$, which is not fully visible to the agent, includes the precise coordinates of the agent $(x, y)$ and the enemy $(x_e, y_e)$, and any spells in flight. It also contains the enemy's properties, and static environmental information like the full topology of the maze and current conditions (biome, weather, day/night status).

An enemy's configuration is a vector of categorical features

$$E = [f_1, f_2, \ldots, f_k], \quad \text{where } f_i \in \{1, \ldots, V_i\}.$$

One feature, $f_d$, is designated as the *diagnostic* feature, while the remaining $k - 1$ features are distractors.

The action space $\mathcal{A}$ is composed of navigation actions $\{up, down, left, right\}$ and three targeting actions $\{bullet1, bullet2, bullet3\}$. A reward is issued based on the correspondence between the chosen bullet $a_j$ and the value of the diagnostic feature $f_d$. The reward function is defined as

$$R(s,a) = \begin{cases} +1, & \text{if } a = \text{bullet}_j \text{ and } j = f_d(s), \\ -1, & \text{if } a = \text{bullet}_j \text{ and } j \neq f_d(s), \\ 0, & \text{otherwise.} \end{cases}$$

This setup forces the agent to learn a policy $\pi(a_t \mid o_t)$ that correctly maps visual evidence of $f_d$ to the corresponding action, while being invariant to variations in the distractor features.

### A.3  AI Architecture

The agent's policy is parameterized by a convolutional neural network (CNN) that processes the RGB observations ($o_t$, downsampled to $84 \times 84$). The network architecture consists of three convolutional layers (32 8×8 filters with stride 4, 64 4×4 filters with stride 2, and 64 3×3 filters with stride 1), each followed by a Rectified Linear Unit (ReLU) activation function. The flattened output of the final convolutional layer is passed to a fully connected layer with 512 units, which feeds into two separate output heads: a policy head for the action distribution $\pi(a_t \mid o_t)$ and a value head for the state-value estimate $V(o_t)$.

### A.4  Policy and Reward Formulation for Movement Agent

The action space is discrete, comprising four navigational actions. The agent's learning is guided by a hybrid reward function, $R_t$, issued at each timestep to encourage efficient navigation to the target. The total reward is a sum of three components:

$$R_t = R_{\text{prox}} + R_{\text{goal}} + R_{\text{time}}.$$

Where each component is defined as:

- **Proximity Reward ($R_{\text{prox}}$):** A dense shaping reward based on the change in Euclidean distance, $d$, to the target. It provides $+0.02$ for moving closer ($d_t < d_{t-1}$) and $-0.02$ for moving farther away ($d_t > d_{t-1}$).
- **Goal Reward ($R_{\text{goal}}$):** A sparse, terminal reward of $+2.0$ is issued when the agent successfully reaches the enemy's location. This reward is 0 at all non-terminal timesteps.
- **Time Penalty ($R_{\text{time}}$):** A small, constant penalty of $-0.001$ is applied at every timestep to incentivize the agent to complete the task as quickly as possible.

### A.5  Policy and Reward Formulation for the Shooting Agent

The action space is discrete, comprising three targeting actions. The agent's learning is guided by a reward function, $R_t$, issued at each timestep based on the correctness of the chosen action, +1 if correct and -1 if incorrect. The agent can shoot multiple times until it makes the correct hit. The episode ends after the agent successfully hits the enemy.

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

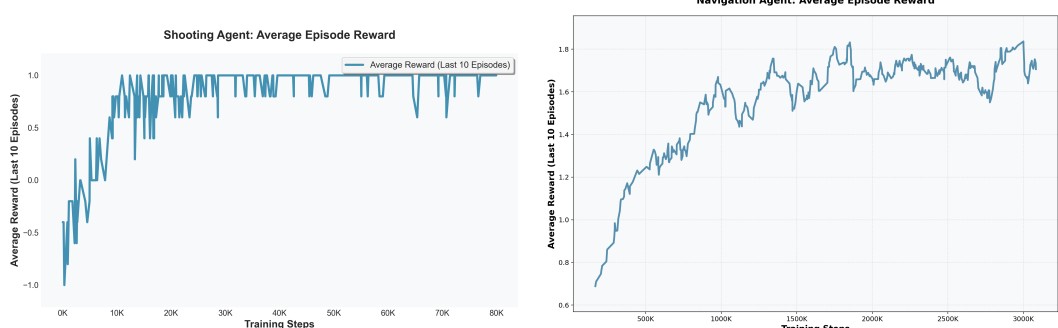

Figure 6: Agent training metrics showing average reward over last 10 episodes (left) and movement behavior during training (right).

Irina Higgins, Loic Matthey, Arka Pal, Christopher Burgess, Xavier Glorot, Matthew Botvinick, Shakir Mohamed, Alexander Lerchner, and Google Deepmind. beta-VAE: Learning Basic Visual Concepts with a Constrained Variational Framework, 2 2017.

Guillaume Lample and Devendra Singh Chaplot. Playing FPS Games with Deep Reinforcement Learning. *31st AAAI Conference on Artificial Intelligence, AAAI 2017*, pages 2140–2146, 9 2016. ISSN 2159-5399. doi: 10.1609/aaai.v31i1.10827. URL `https://arxiv.org/abs/1609.05521v2`.

Valerio Mante, David Sussillo, Krishna V. Shenoy, and William T. Newsome. Context-dependent computation by recurrent dynamics in prefrontal cortex. *Nature 2013 503:7474*, 503(7474):78–84, 11 2013. ISSN 1476-4687. doi: 10.1038/nature12742. URL `https://www.nature.com/articles/nature12742`.

John Schulman, Filip Wolski, Prafulla Dhariwal, Alec Radford, and Oleg Klimov Openai. Proximal Policy Optimization Algorithms. 7 2017. URL `https://arxiv.org/abs/1707.06347v2`.

Larry R. Squire. Memory systems of the brain: a brief history and current perspective. *Neurobiology of learning and memory*, 82(3):171–177, 11 2004. ISSN 1074-7427. doi: 10.1016/J.NLM.2004.06.005. URL `https://pubmed.ncbi.nlm.nih.gov/15464402/`.

Andrea Tacchetti, Leyla Isik, and Tomaso Poggio. Invariant recognition drives neural representations of action sequences. *PLOS Computational Biology*, 13(12):e1005859, 12 2017. ISSN 1553-7358. doi: 10.1371/JOURNAL.PCBI.1005859. URL `https://journals.plos.org/ploscompbiol/article?id=10.1371/journal.pcbi.1005859`.

Naftali Tishby, Fernando C. Pereira, and William Bialek. The information bottleneck method. 4 2000. URL `https://arxiv.org/abs/physics/0004057v1`.

