# OpenReview forum: "Building expertise through task-specific representational alignment in biological and artificial neural networks"
_NeurIPS.cc/2025/Workshop/UniReps — UniReps2025_

### Official Review · Reviewer_CY1h · 2025-09-02
**Good paper but bad fit**

**Confidence:** 4

**Review:**

This work includes well-motivated experiments for measuring the development of expertise in a task across neural systems and the authors provide results that support their hypotheses. However, the authors' understanding of "representational alignment" is different from that of this workshop. They use this phrase to mean internal representations that converge to useful features for task execution and are therefore "aligned" with the task (which just sounds like convergence of a network to me). This misunderstanding is clear as early as the abstract: "Deep reinforcement learning agents playing the game developed representational alignment with the task-relevant features".

At UniReps, "representational alignment" refers to the comparative study of internal representational geometries across different neural systems (artificial vs artificial, biological vs biological, artificial vs artificial). While the "Future Course of Action" section hints at work more in this direction, as it stands, I think this paper would be better suited for a different venue. The observation that task relevant features are best encoded for in the higher levels of networks (both artificial and biological) is not a novel observation in this area of research, and so I don't believe that this is suitable for this workshop. My assessment is less about the quality of the work and much more to do with its fit at UniReps.

**Score:**

1

**Topic Fit:**

1

---

### Official Review · Reviewer_UnQC · 2025-09-11
**A good start to an interesting set of experiments with potential for good discussion, but the core question has been asked before and answered in other contexts. Potentially limiting the novelty of the overall findings.**

**Confidence:** 4

**Review:**

Building expertise through task-specific representational alignment in biological and artificial neural networks review.

In this the authors set out to answer the question of where abstractions arise in deep neural network—biological and artificial, and whether these abstract representations play a causal role in learning i.e. do networks with human-like abstract representations learn to perform certain-tasks better. This question is largely motivated by the observation that many real-world tasks must be performed in myriad environments often with very different observational statistics. These differences in observations distributions are often tangential to the actual task being performed. Thus, in theory a viable strategy for an expert would be to learn to ignore these spurious changes in the environment: becoming invariant to them. Here the authors test whether this happens in practice in CNNs trained to perform a navigation/classification task using RL. The results presented here largely recapitulate known results such as representations becoming increasingly abstract (invariant) in later layers of the network. While the main question of where in deep networks does abstraction arise isn’t really novel, the authors take two interesting spins on it. The first, is by recording neural activity as humans play the same game as the RL agents. The second, is by testing the causal of these abstract representations on learning, by varying the pre-training schemes to get a variety of networks with different levels of abstraction. Both of these are future directions the authors wish to pursue, and should provide useful discussion for people at the workshop.

Questions for the authors:
There seems to be high variance in the performance of your human participants. This variance may make it difficult to align the human representation with those of the agents. Do you have ideas about what the possible sources of this variance could be? For example, is Subject 4 using an improper category boundary or are the deficits here mostly related to the maze navigation aspect of the task? Related, can you query each participant on their inferred boundaries. For example, you can randomly sample 3 enemies using your sampling procedure. Then you can randomly choose one of the 3 to be a reference and ask which of the other 2 are more similar to the reference. This should allow you to estimate a behavioral similarity matrix for each participant which may be useful for diagnosing participant strategies.

In the introduction you briefly mentioned the difference between solutions found by specialist and generalist agents. I’m curious how you think your present results would change if subjects had to perform this task under multiple category boundaries (cued or uncued). This situation would seem more similar to the Mante et al. experiments. In this regime would you expect the final layer to only be invariant over changes in the background in maintaining all information about the enemy features, or would you expect to see context-dependent invariances where you get situation specific invariances but conditioned on a contextual variable that arises earlier in the layer hierarchy? I ask because there is debate regarding the Mante et. al. result about whether the irrelevant stimulus really is pruned out or whether they are maintained but rotated into a subspace of activity orthogonal to the readouts.

**Score:**

4

**Topic Fit:**

2

---

### Official Review · Reviewer_Lisb · 2025-09-12
**Extracting features from a CNN network trained on a Unity simulation**

**Confidence:** 5

**Review:**

The paper raises an interesting and important question: “What transformations do representations undergo in the shift from novice to expert?” It focuses on how the human brain reorganizes its representations during the progression from novice to expert in perceptuo-motor tasks. This question is valuable because it can shed light on which features are emphasized or discarded at different stages of learning. However, in its current form, the submission has several weaknesses, particularly regarding the novelty and depth of the presented results. Below, I outline the main concerns with this study:

**Minor Weaknesses -**
1. The authors don’t mention anything about the human subjects even in the supplementary section. Individual variability is crucial in studies like these. Different people may develop expertise differently. For instance, bottlenecks may or may not generalize well across young vs old human adults. Therefore, it is important to add details about the human subjects.
2. What was the fps of the game in Unity?
3. What does the red vertical line in Figure 2 represents?
4. No details on how the trained models were tested.

**Major Weaknesses -**
1. While the research question is clearly stated, the claims made in the abstract are misleading and extend beyond the scope of this study. For example, on line 13 the authors state that they aim to investigate neural recordings in humans; however, no such data or analysis is actually presented. For a short extended abstract, it is acceptable to focus on a single idea with preliminary results, but the abstract gives readers the false impression that neurophysiological recordings from human subjects are included in the paper. This is also evident on lines 46-50, where the authors claim that “In this study, we investigate the representational bottlenecks that result in task-specific expertise in humans and artificial agents.” However, the study does not provide any such results.
2. The comparison between the behavioral data in Figure 2 and the linear classification accuracy of the CNN architecture does not provide meaningful evidence for the claimed bottlenecks or support the assertions made in the abstract.
3. The authors demonstrate that the network extracts different features across layers. However, this is not a novel finding, as prior work has already established that CNNs exhibit strong architectural biases (e.g., weight sharing) that naturally lead to the learning of simple features such as edge orientations in early layers and more complex features in later layers. The authors show this in their figures but this again does not help in supporting their claims. If the authors would have shown this using a different model architecture such as transformers which lack inductive biases as CNNs, then the results would have made a difference.

The paper would benefit from substantial revisions and the inclusion of results that more directly support its claims. While the underlying idea is interesting and potentially valuable, the results presented in the current form do not demonstrate sufficient novelty.

**Score:**

1

**Topic Fit:**

2

---

### Official Review · Reviewer_akZQ · 2025-09-16
**identifying task-specific nerual code in the visuo-motor hierarchy of human brain**

**Confidence:** 4

**Review:**

The abstract aims to explore an important question about how the neural representation of skills evolves during the practice of a learning task. The authors have developed a visual category learning task in which participants collect information through trial and error to improve their performance. The study involves creating an artificial agent that learns the same task and investigating when, during practice, and where in the brain (using fMRI responses) participants represent the neural code that aligns with the internal representation of that expert artificial agent.
The work is novel, the question is important, and the study is well-designed, and I look forward to seeing the results.

**Score:**

4

**Topic Fit:**

3